# *RAC1*, a Potential Diagnostic and Prognostic Marker for Diffuse Large B Cell Lymphoma

**DOI:** 10.3390/cells11244039

**Published:** 2022-12-14

**Authors:** Xue Wu, Yuan Li, Wandong Zhang, Jing Zhang, Baoan Chen, Zheng Ge

**Affiliations:** 1Department of Hematology, The Affiliated Zhongda Hospital, Institute of Hematology, Southeast University, Nanjing 210009, China; 2Department of Oncology, School of Medicine, Southeast University, Nanjing 210009, China; 3Department of Cellular & Molecular Medicine, Faculty of Medicine, University of Ottawa, Ottawa, ON K1H 8M5, Canada; 4Human Health Therapeutics Research Centre, National Research Council Canada, Ottawa, ON K1A 0R6, Canada

**Keywords:** *RAC 1*, DLBCL, diagnosis, prognosis

## Abstract

The gene changes for diagnosis and prognosis of diffuse large B cell lymphoma (DLBCL) still remain unclear. RAC1 was reported to be asso;ciated with the B cell receptor signal pathway, but its relations with DLBCL have not yet been systematically explored. In this study, we have conducted molecular, bioinformatics and clinical analyses by using publicly available data from The Cancer Genome Atlas (TCGA). Wilcoxon signed-rank test and logistic regression were performed to evaluate the association between RAC1 and clinical features in patients. Kaplan–Meier and Cox regression methods were used to examine the impacts of RAC1 expression level on overall survival, and a nomogram was performed to illustrate the correlation between RAC1 and the risk of DLBCL. Our results revealed that the expression level of RAC1 in DLBCL was higher than that in normal tissues or lymphadenitis. High-level expression of RAC1 was significantly associated with clinical stage, as well as being an independent factor affecting overall survival. RAC1 was negatively correlated with Bruton’s tyrosine kinase (BTK). The association between RAC1 gene expression and the risk of DLBCL was presented in a nomogram. In conclusion, RAC1 expression patterns may be used to predict the development and prognosis of DLBCL.

## 1. Introduction

Diffuse large B cell lymphoma (DLBCL) now is ranked as one of the common cancers, with diversified biological and clinical presentations [1]. The application of the NanoString platform in DLBCL research in recent years provides the possibility of discovering more therapeutic options for the disease, besides the classic CHOP (cyclophosphamide, doxorubicin, vincristine, and prednisone). CHOP-based therapeutic regimens were developed according to the original International Prognostic Index (IPI), described in 1993 [2,3,4]. Therefore, new targets can be explored to improve prognostic prediction and treatment for the disease.

The Rho/Rac family was first identified and studied by Dr. R. Axel’s group in 1985 [5], and functions as a Ras-related gene. The Rho family of GTPase is a family of small (~21 kD) signaling G proteins and a subfamily of the Ras superfamily; while the Rac family is a subfamily of the Rho family of GTPases [6], small (~21 kDa) signaling G proteins (more specifically a GTPase). The Rho/Rac family plays a vital role in the regulation of cell growth, angiogenesis, tissue remodeling and cell migration [7,8]. In most instances, these GTPases convert opportunely between active state and inactive state by the exchange of guanosine diphosphate (GDP) and GTP molecules [9]; this provides a relatively reliable combination with the binding proteins, leading to the activation of the signal cascade. In other cases, Rho/Rac family GTPases also play a role in cancer initiation and progression [10].

The Rho/Rac family can be classified into six subfamilies based on homological and structural characteristics, including CDC42, RAC, RHO, RND, RhoBTB and RhoT/Miro. Among them, the RAC protein is located at a central regulatory point, in terms of lymphomagenesis, in some specific cellular pathways. For example, in adult T cell leukemia/lymphoma (ATL), the activity of *RAC1* is observed to be related to the formation of lamellipodia for the enhancement of tumor cell infiltration [11]. In other cases, such as in mantle cell lymphoma (MCL), over-expression of *RAC1* usually leads to shorter survival and poorer prognosis for patients [12]. It is of note that in the “B cell receptor signal pathway” (https://www.genome.jp/pathway/map04662+K07370, accessed on 27 November 2022), it indicates that there is a relationship between *RAC1*, the VAV family and some upstream tyrosine kinases. This relationship has been already noted in luminal breast cancer [13,14].

Unfortunately, the crucial role of the Rho/Rac family in the development and progression of DLBCL is largely uncharacterized [15]. Our study aims to characterize the role of the Rho/Rac family in DLBCL and to determine whether *RAC1* is a potential diagnostic and prognostic marker for DLBCL.

## 2. Materials and Methods

### 2.1. Multiple Cancer Dataset Source and Preprocessing

The expression levels of Rho/Rac family genes in 48 DLBCL samples were analyzed based on the mRNA sequencing (RNA-seq) dataset of The Cancer Genome Atlas (TCGA-DLBC), with 444 normal samples from the Genotype-Tissue Expression (GTEx) database. Clinical characteristics such as age, gender, clinical stages and primary therapy outcome were obtained, with regard to their corresponding clinical patterns. Additionally, the expression levels of *RAC1* in tumor tissues of 18 cancer types were analyzed and compared to adjacent normal tissues, based on the mRNA sequencing data of TCGA. Differential transcriptional expressions were compared by Student’s *t*-test.

### 2.2. Diagnostic Analysis

A receiver operating characteristic (ROC) curve was applied to assess the specificity and sensitivity of the Rho/Rac family genes’ predictive accuracy, with the area under the ROC curve (AUC) used as the diagnostic value based on the “pROC” package in statistical software (version 3.6.3).

### 2.3. Survival Analysis

The Kaplan–Meier curve was applied to compare the overall survival (OS) between the differential expression groups of Rho/Rac family members, including 48 DLBCL samples in the TCGA database with a mean follow-up of 50, 100, 150 and 200 months, respectively. The correlation between Rho/Rac family members’ expression and survival was analyzed to discover the significant prognostic factors. The hazard ratio (HR) with 95% confidence intervals (CI) and log-rank *p*-value were also computed.

### 2.4. Correlation Analysis

Gene expression correlation analysis was performed for given sets of mRNA expression data in TCGA-DLBC and GTEx. The correlation coefficient was determined by the Spearman method. BTK was used for the x-axis, and other genes of interest were represented on the y-axis. 

### 2.5. Immunohistochemical Staining (IHC)

Firstly, 3% H_2_O_2_ was used to inhibit the endogenous peroxidase activity for 12 min. The epitope retrieval was achieved with 0.1 mol/l citrate buffer at pH 6.0. Sections from paraffin-embedded tissues were then immunostained with a rabbit polyclonal antibody against *RAC1* (dilution: 1:300, Bioss, BJ, CHN). Staining was carried out with 3,3′-diami-nobenzidine (DAB) and then sections were counterstained with haematoxylin. Nonspecific IgG was used as a negative control.

### 2.6. Statistical Analysis

The Wilcoxon rank sum test was used to evaluate the significant differential expression levels of the Rho/Rac family genes in DLBCL and *RAC1* in multiple cancers, with the cut-off value of gene expression selected as the median method. The correlation of gene expression was evaluated by Spearman’s correlation coefficient. Logistic regression was performed to analyze the association between clinical features and *RAC1* expression in DLBCL. Univariate Cox analysis was used to screen potential risk factors, and multivariate Cox analysis was used to verify the independent variate of *RAC1* expression on overall survival. According to the final Cox proportional hazard regression model, and by using the rms package in R version 3.6.3 (http://www.r-project.org/) accessed on 27 November 2022, the nomogram model was constructed to predict 1-, 3- and 5-year DLBCL OS by combining the value of *RAC1* with other clinical variables. We used the calibration analysis to assess the nomogram’s predictive accuracy.

All statistical analyses were performed using R statistical software (version 3.6.3). A *p*-value of less than 0.05 is considered as statistically significant.

## 3. Results

### 3.1. Expression of the Rho/Rac Family Members in DLBCL

The mRNA levels of Rho/Rac family genes in DLBCL were obtained from the RNA-seq datasets of TCGA, as compared to the normal tissue samples. We found that most of the Rho/Rac family members were upregulated in DLBCL. Higher mRNA expression levels of RHOA (*p* = 0.001), RHOC (*p* < 0.001), RHOD (*p* < 0.001), RHOF (*p* < 0.001), RHOJ (*p* < 0.001), RHOQ (*p* = 0.001), RHOU (*p* < 0.001), RHOV (*p* < 0.001), RAC1 (*p* < 0.001), RAC3 (*p* < 0.001), RND1 (*p* < 0.001), RND2 (*p* < 0.001), and RND3 (*p* < 0.001) were observed, while lower mRNA expression levels of RHOB (*p* = 0.002) and RHOG (*p* < 0.001) were seen in DLBCL. The various expression patterns of Rho/Rac family genes in DLBCL, as compared to normal tissues, are illustrated (Figure 1) (Appendix A).

### 3.2. Diagnostic Value of the Rho/Rac Family Members in DLBCL

We used ROC curves to evaluate the Rho/Rac family members’ performance for predicting the outcomes of DLBCL. In predicting the outcomes of normal or tumor tissues, the predictability of the RND/RAC/CDC42 subfamily genes such as RND3 (AUC = 0.968), RHOD (AUC = 0.936), RND2 (AUC = 0.869), RND1 (AUC = 0.840), RHOF (AUC = 0.837), RAC1 (AUC = 0.877), RAC3 (AUC = 0.869), RHOJ (AUC = 0.989) showed a high accuracy, while the RHO subfamily genes such as RHOA (AUC = 0.652), RHOB (AUC = 0.640), RHOC (AUC = 0.666) showed a lower accuracy (Figure 2) (Appendix A).

### 3.3. Prognostic Potential of the Rho/Rac Family Members for DLBCL

For the purpose of determining the prognostic value of the mRNA expression levels of Rho/Rac family genes in DLBCL, Kaplan–Meier curves were used to evaluate the effects of expression levels on patients’ survival and clinical follow-up period. Log-rank tests were conducted to evaluate statistical significance, and all the significant OS curves associated with the Rho/Rac family members were shown (Figure 3). Our analysis revealed that high mRNA expressions of *RAC1* (HR = 0.2, 95% CI: 0.05–0.84, *p* = 0.008) (Figure 3A) and RND1 (hazard ratio [HR] = 0.25, 95% confidence interval [CI]: 0.04–1.45, *p* = 0.022) (Figure 3C) were significantly associated with a better prognosis of OS. In contrast, high mRNA expression levels of RAC2 (HR = 4.50, 95% CI: 0.73–27.87, *p* =0.011) (Figure 3B), CDC42 (HR = 4.83, 95% CI: 0.74–31.36, *p* = 0.007) (Figure 3D), RHOQ (HR = 3.31, 95% CI: 0.77–14.34, *p* = 0.051) (Figure 3E) and RHOF (HR = 3.89, 95% CI: 0.98–15.43, *p* = 0.032) (Figure 3F) were associated with a worse prognosis of OS.

### 3.4. Correlation between the Expression of Rho/Rac Family Members and BTK

To investigate the relationship between the Rho/Rac family members and the BCR signaling pathways, we first analyzed the correlations between BTK and the Rho/Rac family genes by co-expression heatmap. The results indicated that BTK was significantly associated with the expression levels of *RAC1*, RAC2, RHOF, RHOJ, RHOQ, and CDC42 among all the Rho/Rac family members (Figure 4A) (Appendix A). In general, BTK was negatively correlated with *RAC1* (r = −0.350, *p* = 0.014) (Figure 4B), but positively correlated with CDC42 (r = 0.480, *p* = 0.001) (Figure 4C), RAC2 (r = 0.450, *p* = 0.002), RHOF (r = 0.350, *p* = 0.015) (Figure 4D), RHOJ (r = 0.312, *p* = 0.032) and RHOQ (r = 0.340, *p* = 0.018) (Figure 4E).

### 3.5. Expression Levels of RAC1 in Multiple Cancers

To further evaluate the roles of *RAC1* in different types of cancers, we investigated *RAC1* expression by analyzing the RNA-seq data of multiple cancers in TCGA. The results demonstrated that the expression levels of *RAC1* were significantly increased in breast invasive carcinoma (BRCA), cholangiocarcinoma (CHOL), head and neck squamous cell carcinoma (HNSC), kidney renal clear cell carcinoma (KIRC), kidney renal papillary cell carcinoma (KIRP), liver hepatocellular carcinoma (LIHC), lung adenocarcinoma (LUAD), lung squamous cell carcinoma (LUSC) and stomach adenocarcinoma (STAD), compared to corresponding adjacent control samples. However, *RAC1* expression was downregulated in kidney chromophobe (KICH), as compared to the normal controls (Figure 5) (Appendix A). 

### 3.6. Correlation between RAC1 Expression and DLBCL Clinical Characteristics

The correlation identified between *RAC1* expression and clinical characteristics in the patients with DLBCL were summarized in Table 1, including age, gender, clinical stages and primary therapy outcomes. Of the 48 samples, 6 were lacking clinical stage data. The remaining 42 samples included variable stages of DLBCL with differential expression levels of RAC1. Logistic analysis of the association between *RAC1* expression and DLBCL clinical features showed that high-level expression of *RAC1* was significantly associated with clinical stages (III/IV vs. I/II: Odds Ratio [OR] = 0.234, 95% CI = 0.058–0.844, *p* = 0.032) (Table 2).

Analyses of the RNA-seq gene expression data in TCGA revealed that mRNA expression level of *RAC1* was significantly higher in DLBCL stage I/II, as compared to stage III/IV (*p* < 0.05) (Figure 6A) (Appendix A). We have also detected protein expression of *RAC1* in lymphadenitis (Figure 6B) and different clinical stages of DLBCL tissues (Figure 6C–E), by immunohistochemical staining.

### 3.7. High-Level Expression of RAC1 Is an Independent Risk Factor for OS of DLBCL

Univariate Cox analysis showed that high *RAC1* expression was significantly correlated with favor OS (HR = 0.106, 95% CI = 0.013–0.867, *p* = 0.036). In contrast, the primary therapy outcomes, such as progressive disease (PD) and stable disease (SD), were significantly related to poor OS (HR = 0.106, 95% CI = 0.013–0.867, *p* = 0.036). Multivariate Cox analysis confirmed *RAC1* gene expression was an independent risk factor for OS in patients with DLBCL (HR = 10.480, 95% CI = 2.292–47.922, *p* = 0.002) (Table 3). In addition, primary therapy outcome was another independent risk factor in DLBCL (HR = 15.676, 95% CI = 2.603–94.406, *p* = 0.003). Finally, we have constructed a nomogram to predict the 1-, 3-, and 5-year survival probability of the patients, by combining the expression level of *RAC1* with clinical variables. We used the calibration analysis to assess the nomogram’s predictive accuracy: concordance = 0.839 (*p* = 5 × 10^−5^) (Figure 7).

## 4. Discussion

In this study, we have identified the correlation between the different expression patterns of the six subfamilies of the Rho/Rac family, with the initiation, diagnosis and prognosis of DLBCL. The results showed that the expression levels of RHOA, RHOC, RHOD, RHOF, RHOJ, RHOQ, RHOU, RHOV, *RAC1*, RAC3, RND1, RND2 and RND3 were significantly higher in the DLBCL tissues, as compared to the normal tissues; however, the expression levels of RHOB and RHOG stood on the opposite side, indicating that the RHO, RAC and RND subfamilies were associated with the development of DLBCL. Further data analysis found that the expression levels of four subfamilies named RHO, RAC, RND and CDC42 might be meaningful diagnostic markers. Among them, RND2, RND3, RHOD, RAC1, RAC3, RHOJ were the markers which were most reliable for the presence of DLBCL. Meanwhile, analysis for disease prognosis suggests that patients who had a high expression level of RAC2, RND1, CDC42, RHOQ, and RHOF, but a low expression level of *RAC1*, may have a shorter survival time than those who did not. Worth mentioning is that *RAC1* may play roles in the initiation, diagnosis and prognosis of DLBCL all at the same time, based on our analyses.

In fact, as a small GTPase, *RAC1* has an effect on cell behaviors [15,16,17]. It exists not only at the plasma membrane, but also in the nucleus and even in the mitochondria. In 2003, scientists observed that *RAC1* is regulated by multiple receptors, especially by B cell antigen receptor (BCR), resulting in the signal transmission for cell survival, proliferation and differentiation [18]. Over the years, many studies have yielded detailed knowledge on the process of the BCR signaling pathway; it is triggered by a calcium signaling module after antigen binding, leading to BTK activation, followed by the phosphorylated LYN, hereby enabling the downstream activation of the guanine nucleotide exchange factor VAV [19]. VAV binds and activates RAC, having various effects on cell growth [20]. The results in this study inferred that there might be other unknown signaling pathways between BTK and the Rho/Rac family. When investigating the interactions between VAV and RAC, more attention should be paid to the sub-types of them, to explore their effects on each other. Further investigations remain to be conducted on the Rho/Rac family and BTK signaling.

Despite DLBCL, *RAC1* expression is also found in various cancers such as breast invasive carcinoma, gastric malignancies, and lung squamous cell carcinoma. In 2017, Zhang et al. demonstrated that the WNT/PCP–RAC1–JNK signaling pathway might be responsible for breast cancer metastasis organotropism [15]. Two years later, Zhu et al. revealed that by activating *RAC1* and CDC42, gastric cancer cell growth and progression was promoted via MAPK/ERK, PI3K/AKT and PTEN signaling, simultaneously [16]. Furthermore, Yang et al. provided the evidence that *RAC1* is a target of long coding RNAs and microRNAs for the progression of lung cancer, suggesting that *RAC1* may be a new target for drug development for cancer patients with high expression levels of *RAC1* [17]. Overall, our study suggests that *RAC1* occupies a key position in a signaling pathway by which lymphoma cells grow, migrate and die. Furthermore, when *RAC1* expression was evaluated for prognosis, the results showed that Ann Arbor Stage I to II lymphoma patients—who would have better survival—had higher expression levels of *RAC1* than those patients who were defined to develop to stage III to IV, thus indicating that DLBCL patients with higher *RAC1* expression lived longer. The detailed mechanism remains to be further investigated.

Our results also showed that *RAC1* gene expression was higher in breast invasive carcinoma, cholangiocarcinoma, head and neck squamous cell carcinoma, kidney renal clear cell and papillary cell carcinoma, liver hepatocellular carcinoma, lung squamous cell carcinoma, and stomach adenocarcinoma, as compared to corresponding adjacent normal tissues. In particular, *RAC1* expression level was higher in DLBCL, compared to normal lymph tissue or lymphadenitis, and more importantly it was correlated to DLBCL clinical stages. Furthermore, we used ROC analysis to verify the potential prognostic value of *RAC1* in DLBCL. Kaplan–Meier survival analysis revealed that patients with a low expression of *RAC1* had shorter OS (*p* = 0.008). Multivariate Cox analysis further confirmed that low *RAC1* expression was an independent risk factor for OS in patients with DLBCL. At present, a predictive nomogram for DLBCL, by combining the expression value of *RAC1* with clinical variables, has not been reported in the literature. Therefore, we generated a prognostic nomogram by integrating clinical primary therapy outcomes and *RAC1* gene expression via TCGA dataset, to predict the risk of individual patient morbidity and to guide precise individual therapeutic decisions.

In summary, this study indicates that *RAC1* overexpression may not only serve as a predictive factor for the development of DLBCL, but also serve as a prognostic marker for DLBCL. This notion was further supported by the results from IHC analysis for the correlation of *RAC1*, with different clinical stages of DLBCL. Our study also suggests that *RAC1* may serve as a potential target for the development of therapeutic approaches for DLBCL. However, the signaling pathway(s) involved remain to be further investigated.

## Figures and Tables

**Figure 1 cells-11-04039-f001:**
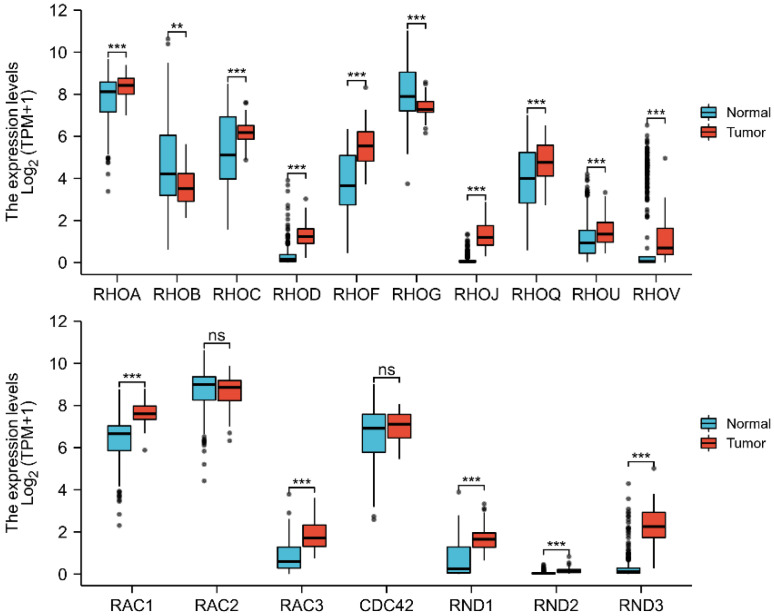
Evaluation of Rho/Rac family genes expression in patients with DLBCL as compared to the normal tissue samples. ** *p* < 0.01; *** *p* < 0.001; ns: non-significance.

**Figure 2 cells-11-04039-f002:**
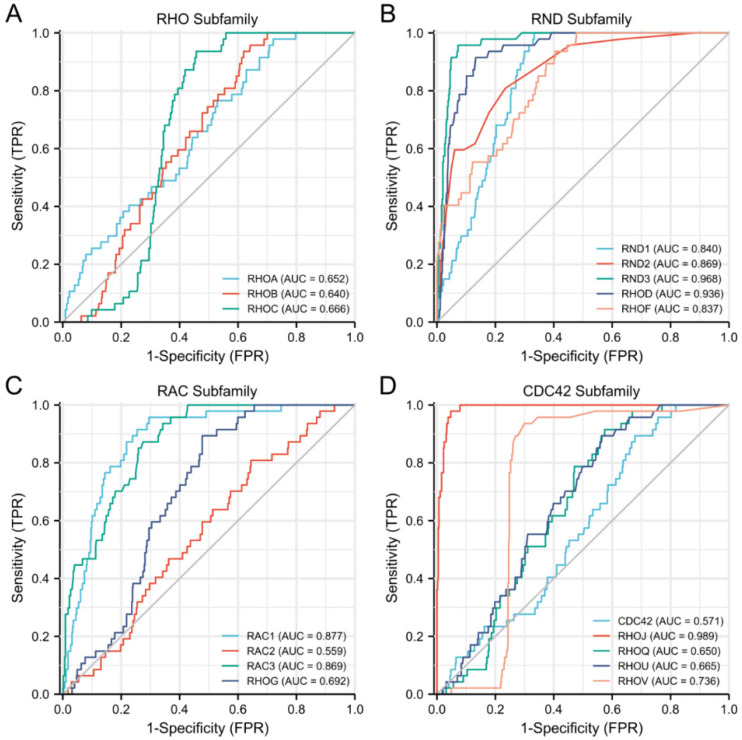
ROC curves and the predictability of Rho/Rac family genes in DLBCL diagnosis. (**A**) RHO subfamily; (**B**) RND subfamily; (**C**) RAC subfamily; (**D**) CDC42 subfamily; ROC, receiver operating characteristic.

**Figure 3 cells-11-04039-f003:**
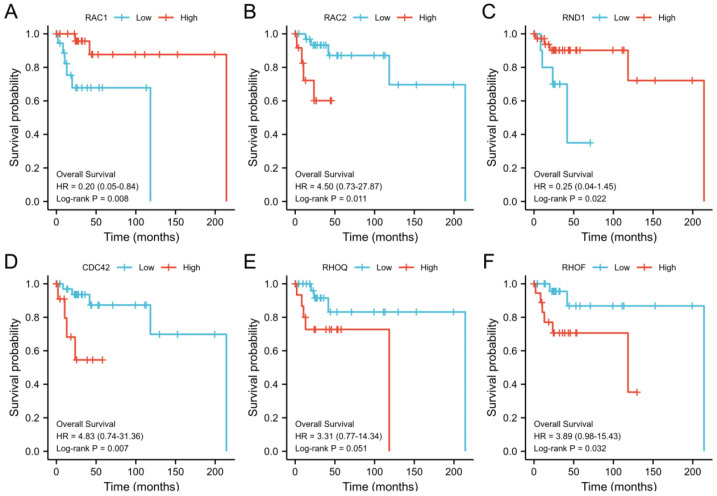
Overall survival and the expression levels of Rho/Rac family genes in DLBCL. Kaplan–Meier curve of overall survival for individual member (**A**) RAC1, (**B**) RAC2, (**C**) RND1, (**D**) CDC42, (**E**) RHOQ, (**F**) RHOF; HR, hazard ratio.

**Figure 4 cells-11-04039-f004:**
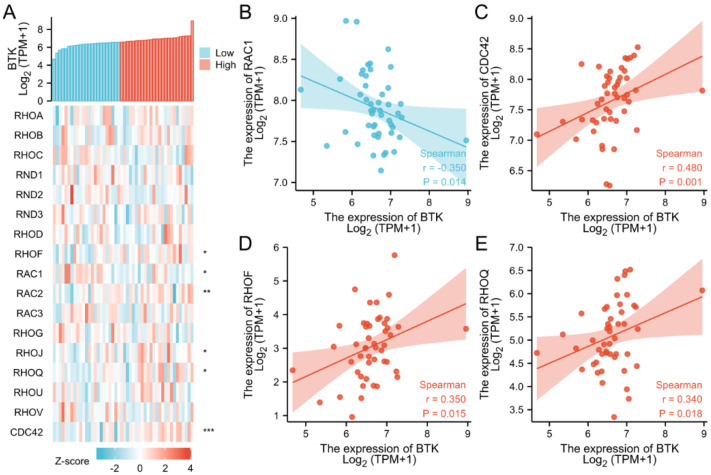
Correlation between the expression of Rho/Rac family members’ and BTK. (**A**) Co-expression heatmap of Rho/Rac family genes and BTK; (**B**) Scatter diagram demonstrated the correlation between BTK and *RAC1*, (**C**) CDC42, (**D**) RHOF and (**E**) RHOQ. * *p* <0.05; ** *p* < 0.01; *** *p* < 0.001.

**Figure 5 cells-11-04039-f005:**
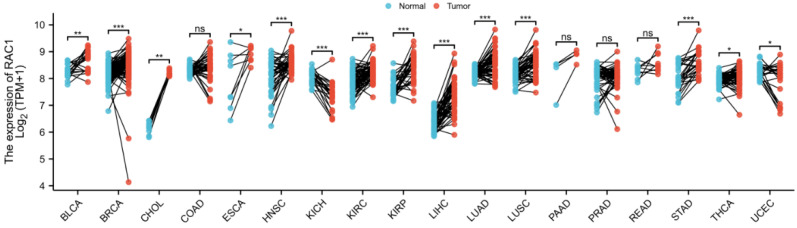
*RAC1* expression levels in 18 different types of tumor tissues and paired adjacent normal tissues. * *p* < 0.05; ** *p* < 0.01; *** *p* < 0.001; ns: non-significance. BLCA, Bladder Urothelial Carcinoma; BRCA, Breast invasive carcinoma; CHOL, Cholangio carcinoma; COAD, Colon adenocarcinoma; ESCA, Esophageal carcinoma; HNSC, Head and Neck squamous cell carcinoma; KICH, Kidney Chromophobe; KIRC, Kidney renal clear cell carcinoma; KIRP, Kidney renal papillary cell carcinoma; LIHC, Liver hepatocellular carcinoma; LUAD, Lung adenocarcinoma; LUSC, Lung squamous cell carcinoma; PAAD, Pancreatic adenocarcinoma; PRAD, Prostate adenocarcinoma; READ, Rectum adenocarcinoma; STAD, Stomach adenocarcinoma; THCA, Thyroid carcinoma; UCEC, Uterine Corpus Endometrial Carcinoma.

**Figure 6 cells-11-04039-f006:**
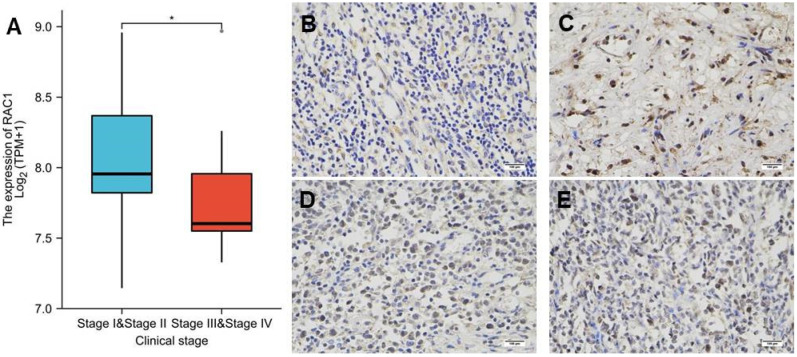
*RAC1* expression in patients with DLBCL according to different clinical stages.(**A**) In the TCGA-DLBC dataset, *RAC1* was upregulated in stage I/II as compared with that in stage III/IV. (**B**) *RAC1* protein IHC staining in lymphadenitis and (**C**) DLBCL stage II, (**D**) stage III, (**E**) stage IV. Magnification 20 × 20, * *p* < 0.05. The staining of 3,3′-diaminobenzidine (DAB) was employed for *RAC1* expression (yellow and brown) compared to negative sample (blue). Scale bar (bottom, right) = 100 μm.

**Figure 7 cells-11-04039-f007:**
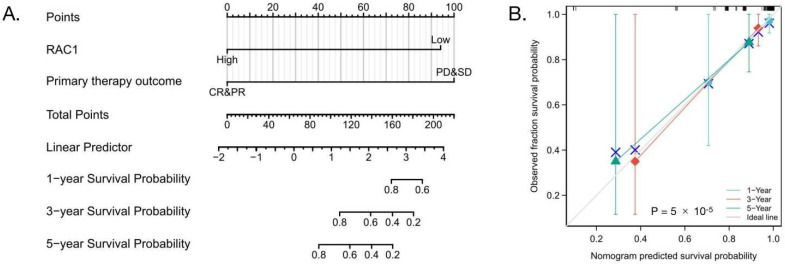
(**A**) Nomogram to predict 1-, 3-, and 5-year OS for DLBCL patients. (**B**) The calibration curves for predicting patient OS at 1-, 3-, and 5-year; the OS predicted by the nomogram model is plotted on the *x*-axis, and the actual OS is plotted on the *y*-axis. OS, overall survival.

**Table 1 cells-11-04039-t001:** Correlation between *RAC1* expression and DLBCL clinical characteristics.

Characteristic	Low-Level Expression of *RAC1*	High-Level Expression of *RAC1*	*p*
n	24	24	
Age, n (%)			0.244
≤60	11 (22.9%)	16 (33.3%)	
>60	13 (27.1%)	8 (16.7%)	
Gender, n (%)			0.772
Female	12 (25%)	14 (29.2%)	
Male	12 (25%)	10 (20.8%)	
Clinical stage, n (%)			0.056
Stage I	1 (2.4%)	7 (16.7%)	
Stage II	8 (19%)	9 (21.4%)	
Stage III	4 (9.5%)	1 (2.4%)	
Stage IV	8 (19%)	4 (9.5%)	
Primary therapy outcome, n (%)			0.26
PD	3 (6.5%)	2 (4.3%)	
SD	2 (4.3%)	1 (2.2%)	
PR	3 (6.5%)	0 (0%)	
CR	15 (32.6%)	20 (43.5%)	
Age, mean ± SD	56.67 ± 14.42	55.88 ± 13.76	0.847

PD, progressive disease; SD, stable disease; PR, partial response; CR, complete response.

**Table 2 cells-11-04039-t002:** Logistic analyses of *RAC1* expression and clinical characteristics of DLBCL.

Characteristics	Total (N)	OR	*p* Value
Age (>60 vs. ≤60)	48	0.423 (0.127–1.338)	0.149
Gender (Male vs. Female)	48	0.714 (0.225–2.229)	0.563
Clinical stage (Stage III and Stage IV vs. Stage I and Stage II)	42	0.234 (0.058–0.844)	0.032
Primary therapy outcome (PD and SD vs. CR and PR)	46	0.540 (0.099–2.520)	0.441

OR: Odds Ratio.

**Table 3 cells-11-04039-t003:** Cox regression analyses of clinical features associated with DLBCL overall survival.

Characteristics	Total (N)	Univariate Analysis		Multivariate Analysis	
		HR (95% CI)	*p* Value	HR (95% CI)	*p* Value
Age	48				
≤60	27	Reference			
>60	21	1.666 (0.416–6.673)	0.471		
Gender	48				
Female	26	Reference			
Male	22	1.039 (0.250–4.324)	0.958		
Clinical stage	42				
Stage I and Stage II	25	Reference			
Stage III and Stage IV	17	1.917 (0.422–8.715)	0.400		
Primary therapy outcome	46				
CR and PR	38	Reference			
SD and PD	8	10.480 (2.292–47.922)	**0.002**	15.676 (2.603–94.406)	**0.003**
*RAC1*	48				
Low	24	Reference			
High	24	0.106 (0.013–0.867)	**0.036**	0.075 (0.008–0.726)	**0.025**

HR, hazard ratio; CI, confidence interval.

## Data Availability

Data are contained within the article or Appendix A.

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
