# Peer review of "RAC1, a Potential Diagnostic and Prognostic Marker for Diffuse Large B Cell Lymphoma"

_cells, 2022, doi:10.3390/cells11244039_

Round 1

Reviewer 1 Report

In this manuscript, the authors conduct molecular, bioinformatics and clinical analyses on publicly available data from The Cancer Genome Atlas (TCGA) to evaluate the association between Rho/Rac family members and clinical features in DLBCL patients. They identify RAC1 expression as correlating with clinical stage and overall survival. 

Overall, the data presented are novel, but improvements need to be made before the manuscript is accepted for publication. English editing is required to make the language clearer and facilitate the reader’s understanding of the data and some methods are lacking. Specific comments are:

Lines 188-190

« High-level expression of RAC1 was significantly correlated with clinical stages in a total of 48 patients with 24 showing a high- and 24 having a low-level expression of RAC1. »

. Unclear what this sentence means.

- Methods state that 32 additional cancer types from TCGA were analysed for RAC1 expression in tumour versus normal samples. In figure 5, only 18 cancer types are shown.

- Legend of figure 5 should include meaning of abbreviated cancer types.

- Were the samples shown in Figure 5 matched tumour/normal tissues? If so, this should be stated in the methods and figure legend.

Lines 188-189 state that RAC1 expression correlates with clinical stage but in Table 1 , expression of RAC1 does not significantly correlate with clinical stages (p =0.056). 

- Why were only 42 of the 48 patients analysed for clinical stage comparisons in Table 2?

Lines 249-253. Data presented in manuscript do not provide evidence that BTK acts as an activator nor an inhibitor of Rho/Rac family members. These sentences should be removed.

- Lines 266-269: data presented in study cannot be used to infer that “RAC1 occupies a key position in several signaling pathways (not simply an axis, but a network)…” These sentences should be removed.

- Methods are missing describing how the predictive nomogram was built, including assessment of the nomogram’s predictive accuracy, cross-validation within the same patient cohort e.g. by leave-one-out method, and external validation on an independent patient cohort to determine if the predictive value of the nomogram remains valid for other DLBCL datatset(s).

Reviewer 2 Report

Original research, the manuscript is well written 

In this paper, the Authors conducted a bioinformatic and clinical analysis to evaluate the association between the expression level of GTPase protein RAC1 and diffuse large B cell lymphoma (DLBCL) prognosis. The Authors used publicly available data from The Cancer Genome Atlas (TCGA) to detect the expression of Rho/Rac family genes in 48 DLBCL samples, finding out the upregulation of most of them in DLBCL. By ROC curves, they assessed the diagnostic accuracy for predicting tumor tissues outcome. By Kaplan Meier curves, they observed a better DLBCL prognosis of overall survival in patients with high mRNA expression of RAC1 and RND1. Through co-expression heatmap, they also investigated the correlation between the expression of Rho/Rac family members and Bruton’s tyrosine kinase (BTK), involved in B cell development, concluding that BTK was an activator for most Rho/Rac family members and an inhibitor only for RAC1.

Moreover, they used univariate and multivariate analysis to compare RAC1 expression levels in DLBCL with multiple cancers: unlike most carcinomas, they discovered that the high expression of RAC1 was related to better DLBCL overall survival. Finally, analysing RNA-seq gene expression data in TCGA and after immunohistochemical staining (IHC), they could observe a higher expression of RAC1 in the first and second DLBCL tumor stages compared to normal tissue and lymphadenitis. This data suggested the involvement of a new RAC1 pathway different from the canonical one.

In my opinion, the study is precise and accurate from a bioinformatic and statistical point of view. The aim of this work is well explained. I recommend adding a more detailed caption for IHC analysis figure (figure 6 B-E), in which highlighting the staining of RAC1 protein and negative control makes it easier to read and understand; as is the image is not clear.

Reviewer 3 Report

The study concerns the evalutation of the the Rho/Rac family members and the 157 BCR signaling pathways, in normal tissue and in DLBCL . A correlation between Rho/Rac family genes and BTK was also analysed. Some Rho/Rac genes a resulted overexpressed in DLBCL and  also correlated with clinical caharacteritics at diagnosis and with  prognosis,. Some other genesof the Rho/rac superfamily are associated with BTK and  thus their expression has  therapeutical implication .

The study is of of interest,  well written and te m biological methid compltely explained .

A more extensive correlation with clincal characteristics (at least those inclueìde in the IPI ) and immunohistochemical/cytogentic feaures (i.e. expression of myc, bcl2 , bcl6 and so on)  features  should be implemented to better underline the clinical implication of the findings

Furthermore, the treatment choice and the relapse  rate are completely lacking and it would be of great interest showing these data  

Round 2

Reviewer 1 Report

The Authors have adequately addressed the comments made. Perhaps the Authors could also include the plot showing the nomogram predicted survival probability versus observed fraction survival probability (shown at the end of the cover letter) in the manuscript.
